# Differentially Expressed Genes and Signaling Pathways Potentially Involved in Primary Resistance to Chemo-Immunotherapy in Advanced-Stage Gastric Cancer Patients

**DOI:** 10.3390/ijms24010001

**Published:** 2022-12-20

**Authors:** Mauricio P. Pinto, Matías Muñoz-Medel, Ignacio N. Retamal, MariaLoreto Bravo, Verónica Latapiat, Miguel Córdova-Delgado, Charlotte N. Hill, M. Fernanda Fernández, Carolina Sánchez, Mauricio A. Sáez, Alberto J. M. Martin, Sebastián Morales-Pison, Ricardo Fernandez-Ramires, Benjamín García-Bloj, Gareth I. Owen, Marcelo Garrido

**Affiliations:** 1Department of Hematology and Oncology, Faculty of Medicine, Pontificia Universidad Católica de Chile, Santiago 8330032, Chile; 2Masters’ Program of Research in Health Sciences, School of Medicine, Pontificia Universidad Católica de Chile, Santiago 8330077, Chile; 3Fundación GIST Chile, Santiago 7550387, Chile; 4Centro de Oncología de Precisión, Escuela de Medicina, Universidad Mayor, Santiago 7560908, Chile; 5Programa de Doctorado en Genómica Integrativa, Vicerrectoría de Investigación, Universidad Mayor, Santiago 7500994, Chile; 6Laboratorio de Redes Biológicas, Centro Científico y Tecnológico de Excelencia Ciencia & Vida, Fundación Ciencia & Vida, Santiago 7780272, Chile; 7Escuela de Ingeniería, Facultad de Ingeniería, Arquitectura y Diseño, Universidad San Sebastián, Santiago 8420524, Chile; 8Facultad de Medicina y Ciencias de la Salud, Universidad Mayor, Santiago 7510041, Chile; 9Faculty of Biological Sciences, Faculty of Medicine, Pontificia Universidad Católica de Chile, Santiago 8320000, Chile; 10Millennium Institute on Immunology and Immunotherapy, Santiago 8331150, Chile; 11Advanced Center for Chronic Diseases, Santiago 8380492, Chile

**Keywords:** molecular oncology, cancer therapy, gastric cancer, immunotherapy resistance, angiogenesis, Wnt pathway, β-catenin, PI3K pathway

## Abstract

Recently, the combination of chemotherapy plus nivolumab (chemo-immunotherapy) has become the standard of care for advanced-stage gastric cancer (GC) patients. However, despite its efficacy, up to 40% of patients do not respond to these treatments. Our study sought to identify variations in gene expression associated with primary resistance to chemo-immunotherapy. Diagnostic endoscopic biopsies were retrospectively obtained from advanced GC patients previously categorized as responders (R) or non-responders (NR). Thirty-four tumor biopsies (R: n = 16, NR: n = 18) were analyzed by 3′ massive analysis of cDNA ends (3′MACE). We found >30 differentially expressed genes between R and NRs. Subsequent pathway enrichment analyses demonstrated that angiogenesis and the Wnt-β-catenin signaling pathway were enriched in NRs. Concomitantly, we performed next generation sequencing (NGS) analyses in a subset of four NR patients that confirmed alterations in genes that belonged to the Wnt/β-catenin and the phosphoinositide 3-kinase (PI3K) pathways. We speculate that angiogenesis, the Wnt, and the PI3K pathways might offer actionable targets. We also discuss therapeutic alternatives for chemo-immunotherapy-resistant advanced-stage GC patients.

## 1. Introduction

Global reports indicate that gastric cancer (GC) is the fifth most common malignancy and the fourth leading cause of cancer death [1]. Regarding incidence and mortality, studies demonstrate that GC incidence is higher among males, and mortality is higher in specific geographic areas such as Eastern Asia and South America [2]. As such, GC is the second leading cause of cancer death in Chile, claiming > 3000 lives/year. High mortality rates are at least partially explained by late diagnosis. Indeed, our research group recently reported that >60% of newly diagnosed GC cases were advanced stage in a cohort of Chilean patients [3]. For decades, the standard treatment for advanced-stage GC consisted in a combination of fluoropyrimidines and platinum compounds (hereafter simply called chemotherapy). Unfortunately, overall survival (OS) using chemotherapy was typically <1 year.

On June 2021, the CheckMate 649 trial demonstrated that the addition of nivolumab (a form of immunotherapy) to chemotherapy increased OS to >14 months [4]. Nivolumab is a humanized monoclonal antibody against programmed cell death-1 protein (PD1). Tumor cells can express programmed death ligand-1 (PDL1) that binds and activates PD1 in T-cells. Upon binding, activated PD1 causes T-cell “exhaustion”, a process that impairs the antitumoral activity of T-cells and favors tumor progression. Therefore, the use of nivolumab prevents PDL1-PD1 binding and increases antitumoral T-cell activity, which launches an immune attack on cancer cells. Given its positive results, the chemo-immunotherapy combination became the new standard of care for advanced-stage GC. However, despite its efficacy, the same CheckMate 649 indicated that about 40% of patients experience little or no benefit derived from this treatment.

Investigators have previously defined eight emerging “hallmarks of resistance” to immunotherapy or immune-resistance nodes that should guide future research on this topic [5]. Studies have identified various mechanisms explaining primary and acquired resistance to immunotherapy [6]. Primary resistance occurs when patients do not respond to initial therapy. This phenomenon is driven by intrinsic tumor factors such as altered cell signaling and the tumor microenvironment (TME). Essential cell signaling pathways associated with immunotherapy resistance in GC include the phosphoinositide 3-kinase (PI3K)-Akt-mammalian target of rapamycin (mTOR) pathway, the mitogen-activated protein kinase (MAPK) pathway, and the Wnt/β-catenin pathway. First, the PI3K-Akt-mTOR pathway is altered in >70% of esophageal carcinomas [7]. Studies in lung cancer demonstrate that PI3K activation increases PDL1 expression [8]. Conversely, PI3K inhibition enhances CD8^+^T-cell tumor infiltration. In melanoma, phosphatase and tensin homolog (*PTEN*) loss is associated with immunotherapy resistance; since PTEN is a negative regulator of PI3K these alterations can lead to increased PI3K activity [9]. Secondly, the MAPK is up-regulated in 50–60% of GCs [10]. This pathway involves activation of the Ras/Raf MAPK and the ERK kinase. In malignant cells, MAPK activation increases vascular endothelial growth factor (VEGF) secretion, a potent proangiogenic. Studies demonstrate that VEGF is also an immunosuppressor that increases the recruitment of regulatory T-cells (Tregs) and impairs T-cell function and recruitment and the differentiation and activation of antigen-presenting (dendritic) cells. Third, the Wnt/β-catenin signaling pathway plays a pivotal role in several physiological processes, including proliferation, survival, cell death, and tissue homeostasis in adult individuals. Aberrant activation of Wnt/β-catenin is reported in 30–50% of GCs [11]. Studies based on The Cancer Genome Atlas (TCGA) demonstrate that Wnt/β-catenin mutations are associated with non-T-cell-inflamed, tumors which are characterized by poor infiltration of anti-tumoral T-cells and natural killer (NK) cells [12]. Other mechanisms of primary resistance include alterations in the interferon gamma (IFNγ) or the transforming growth factor-β (TGFβ) pathway.

Unlike primary resistance, secondary (or acquired) resistance occurs when patients display an initial favorable response to chemo-immunotherapy. However, after a period, tumors continue their progression. This is generally due to impaired or loss of T-cell function that decreases antigen presentation and IFNγ signaling. Other alternatives include β2-microglobulin (β2M) and Janus activated kinases (JAK) 1/2 mutations [6].

Herein, we sought to identify up-regulated and down-regulated genes associated with primary resistance to chemo-immunotherapy in a cohort of GC patients previously characterized as responders (R) or non-responders (NR).

## 2. Results

### 2.1. Selection of Patients

Our study sought to identify differentially expressed genes comparing chemo-immunotherapy responders (R) to non-responders (NR). First, we selected a group of 39 advanced GC patients that had undergone chemo-immunotherapy treatments and were categorized as R or NR. Then, we retrospectively collected endoscopic biopsies obtained at the time of diagnosis and, therefore, could be considered as treatment naïve. We also collected basic clinical information from medical records. This is summarized in Table 1. Briefly, the median age at diagnosis was 61 years old, and patients were predominantly males (76.9%) and of Lauren intestinal type (59%). Overall, 20 out of 39 patients were chemo-immunotherapy Rs, whereas 19 were NRs. As expected, Kaplan–Meier curves showed that median OS was significantly higher on Rs (21 months) versus NRs (13 months; Figure 1a; log rank *p* = 0.018).

### 2.2. Analysis of Differentially Expressed Genes between R and NR

Next, we sought to determine which genes were differentially expressed by comparing R versus NRs and analyzed gene expression levels using the 3′ massive analysis of cDNA ends (3′MACE) technique. At this stage, five samples were excluded due to incomplete datasets or by quality controls, and, therefore, subsequent analyses included 34 patient samples (Rs: n = 16, and NRs: n = 18). This technique captures messenger RNA (mRNA) from samples and provides partial sequences (3′ cDNA ends) of enriched transcripts to quantify their abundance. Figure 1b shows an unsupervised clustering analysis of differentially expressed (up- and down-regulated) genes in our 34 samples. Note that samples obtained from Rs and NRs tend to group separately. The lower panel also includes clinical characteristics of patients at diagnosis. As pointed out, most patients were males; diffuse Lauren type and signet-ring cell positivity were more frequent among NRs. Figure 1c shows a waterfall plot of down-regulated genes (8 genes; red) or up-regulated genes in NRs (28 genes; blue).

**Figure 1 ijms-24-00001-f001:**
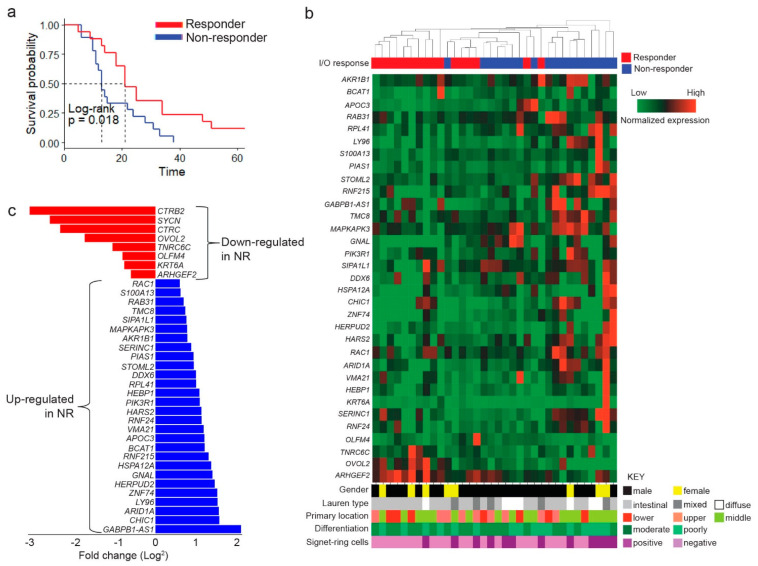
Differentially expressed genes in chemo-immunotherapy responders and non-responders by 3′ MACE. (**a**) Kaplan-Meier curve comparing survival rates between advanced GC patients categorized as responders (R) or non-responders (NR) to chemo-immunotherapy (LogRank, *p* = 0.018). (**b**) Unsupervised clustering of differentially expressed genes analyzed by 3′ massive analysis of cDNA ends (3′MACE). The lower panel shows clinical characteristics of patients (**c**) Waterfall plot of differentially expressed genes down-regulated or up-regulated in NR patients.

### 2.3. Analysis of Enriched Pathways and NGS in NR Patients

Subsequently, after obtaining the lists of differentially expressed genes, we sought to determine which were specifically enriched in NR. We selected a total of 33 differentially expressed genes contained in the PANTHER database and searched for enriched signaling pathways. As shown in Figure 2a, the most enriched pathways were angiogenesis (three genes) and the Wnt signaling pathway (three genes), followed by the histamine H2-mediated signaling pathway (one gene) and the β3-adrenergic receptor signaling pathway (one gene). To partially confirm these findings, we applied a less stringent cutoff (*p* < 0.001) to increase the number of differentially expressed genes and discover other potentially enriched pathways. As a result, we confirmed angiogenesis and the Wnt signaling pathways as highly enriched (three genes each). Additionally, we found the RAS pathway (two genes), the p38 MAPK pathway (two genes), the VEGF signaling pathways (two genes), and the plasminogen-activating cascade (one gene). These results are shown in Appendix A. It is noteworthy that all genes in the angiogenesis and the Wnt signaling pathways were up-regulated in NRs.

Previous studies have demonstrated that the PI3K pathway is frequently altered in GC [13]. Furthermore, this pathway plays a central role in angiogenesis, and studies have documented a PI3K/Wnt signaling crosstalk [14]. Therefore, next we sought to confirm (at least partially) our findings and analyzed a subset of four NR patients (chemo-immunotherapy) using NGS. Figure 2b summarizes relevant alterations in *TP53*, *PIK3CA*, *PIK3R1*, *APC*, *ARID1A*, and *SMAD4*. Note that three out of four NRs displayed *TP53* alterations (SNV, INs, or CNV). Interestingly, all four NR patients harbored alterations in the genes of the PI3K (*PIK3CA*/*PIK3R1*) or the Wnt signaling pathway (*APC*/*ARID1A*). A more detailed description of alterations is summarized in Appendix A.

## 3. Discussion

Despite the undeniable success of immune checkpoint inhibitors (ICIs, such as nivolumab) in the clinic, a substantial fraction of cancer patients remain unresponsive to these treatments [15] even when they are used in combination with chemotherapeutic agents [16]. In advanced-stage GC, this subset can be up to 40% of cases [4]. Several studies have searched and postulated various primary and secondary resistance mechanisms to ICIs. The current consensus indicates that immediate resistance to ICIs results from altered intracellular signaling in tumor cells and its interaction with the TME. Although we could not characterize the TME of patients in our study, we performed pathway enrichment analyses that included differentially expressed genes in chemo-immunotherapy R and NR samples. Our results suggest that the angiogenesis pathway and the Wnt signaling pathways are up-regulated in NRs (Figure 2a,b). Previous studies have demonstrated that tumor angiogenesis has an immunosuppressive effect on the TME. This occurs by direct inhibition of antigen-presenting and immune effector cells and by potentiating the activity of immune suppressor cells such as Tregs and TAMs via secretion of VEGF and angiopoietin-2 [17]. Similarly, the Wnt signaling pathway plays a pivotal role in immune exclusion and tumor resistance against immune attacks [18,19]. As shown in Figure 3, activation of this pathway is mediated by cytoplasmic accumulation of β-catenin that acts as a transcription cofactor controlling Wnt-regulated genes. A recent review by Li et al. [20] suggests that Wnt pathway mediated immune exclusion is achieved by at least three mechanisms: (1) induction of the transcriptional repressor activating transcription factor-3 (ATF3) in dendritic cells, (2) up-regulation of Treg survival, and (3) activation of TAMs to secrete interleukin 1-beta (IL-1β) and interleukin-6 (IL-6). Firstly, aberrant Wnt signaling triggers the induction of ATF3 in dendritic cells, inhibiting the production of the chemokine CCL4 [21]. Since CCL4 is a chemoattractant for the recruitment of immune cells with potential anti-tumoral activity such as NK and “naïve” T-cells, the decrease in CCL4 translates into a reduction of T-cell proliferation, activation, and infiltration into the tumor. Secondly, empirical studies have suggested that aberrant Wnt signaling increases Treg survival. However, the precise mechanism for this interaction is still unknown [22]. Thirdly, tumor-associated macrophages (TAMs) within the TME can become activated by cancer cells via an unspecified mechanism, presumably related to Snail stabilization and the epithelial-to-mesenchymal transition (EMT) that triggers the secretion of IL-1β and IL-6. Subsequently, IL1-β secreted by TAMs stabilizes intracellular β-catenin potentiating Wnt signaling in cancer cells via NFkB/PDK1 [23] and glycogen synthase kinase 3-beta (GSK3β) inhibition, whereas IL-6 further amplifies inflammation within the TME. These three mechanisms are incorporated into the hypothetical model shown in Figure 3. We speculate that Wnt/β-catenin pathway activation in chemo-immunotherapy-resistant tumors can be achieved directly via mutations on *APC* or *CTNNB1*. Indeed, our research group recently reported that these alterations are frequent among Chilean GC patients [3]. Furthermore, *APC* and *CTNNB1* mutations are usually driver mutations in GC. In the first case, APC is part of the β-catenin destruction complex in the cytoplasm. Therefore, the loss of function *APC* mutations increases β-catenin accumulation and Wnt signaling. In the second case, mutations of the *CTNNB1* gene (that encodes β-catenin) can affect phosphorylation sites that are key for protein degradation that results in β-catenin accumulation. Alternatively, mutations that affect the PI3K signaling pathway (either *PIK3CA/PIK3R1* mutations or *PTEN* loss) can also increase the accumulation of cytoplasmic β-catenin via GSK3β inhibition by phosphorylated Akt. Along with APC, GSK3β is part of the β-catenin destruction complex, and its inhibition increases Wnt signaling. Notably, PI3K aberrant activity can also increase angiogenesis via mTOR and VEGF. As explained, VEGF is an immunosuppressor that facilitates Treg recruitment, contributing to immune exclusion.

Based on our findings, we propose potential therapeutic interventions for chemo-immunotherapy-resistant patients. These are summarized in Figure 3 and could target angiogenesis, the PI3K pathway, or the Wnt/β-catenin pathway. For several years, the angiogenesis blockade has been a target of interest in GC [24]. However, initial studies using bevacizumab (anti-VEGF) did not reach statistical significance regarding OS [25]. More recently, preliminary results from the phase II NIVACOR trial demonstrated good tolerance and acceptable toxicity of a triple treatment that combined FOLFOXIRI/bevacizumab and nivolumab as first-line treatment in RAS/BRAF mutant metastatic colorectal cancer patients [26]. This therapy is yet to be tested in advanced GC and will probably require incorporating companion diagnostics to identify those patients more likely to respond. Ramucirumab is another antiangiogenic that targets the VEGF receptor (VEGFR2). Phase III studies in GC patients demonstrated the efficacy of ramucirumab alone or in combination with paclitaxel [27,28]. To date, ramucirumab remains a second-line therapy for advanced-stage GC. Interestingly, a recent phase I/II trial reports promising antitumor activity and manageable toxicity with the combination of nivolumab and paclitaxel plus ramucirumab as a second-line in advanced GC patients [29]. As explained, the PI3K pathway is frequently mutated in GC. Figure 2b shows the three out of four chemo-immunotherapy-resistant patients who displayed clinically relevant mutations in *PIK3CA*/*PIK3R1*. These results suggest PI3K inhibition could be a therapeutic alternative for chemo-immunotherapy-resistant patients; obviously, this should be further validated in a larger cohort. Although several PI3K inhibitors are under development, many have demonstrated their efficacy at the preclinical stage, and a few have been tested in phase I/II trials [30]. Perhaps, the most successful PI3K inhibitor in the clinic is alpelisib. The combination of alpelisib plus fulvestrant (an estrogen receptor antagonist) demonstrated safety and efficacy in hormone receptor positive, HER2-negative, *PIK3CA* mutant advanced breast cancer patients previously treated with cyclin-dependent kinase 4/6 (CDK4/6) inhibitors [31]. In GC patients, an ongoing phase Ib/II trial is recruiting advanced-stage *PIK3CA* mutants to receive a combination of alpelisib plus paclitaxel (ClinicalTrials.gov identifier: NCT04526470).

As explained, the Wnt pathway not only controls several physiological processes but also plays crucial roles in carcinogenesis, cancer progression, and resistance to immunotherapy. Consequently, developing Wnt inhibitors with clinical applicability is an area of active research [32,33]. Unfortunately, as with PI3K inhibitors, most Wnt signaling inhibitors have demonstrated their efficacy in preclinical studies. Figure 3 highlights (green box) three significant categories of Wnt inhibitors under development: porcupine (PORCN) inhibitors, tankyrase (TNKS) inhibitors, and CBP/β-catenin antagonists. Porcupine is a membrane-bound O-acyltransferase; the palmitoylation of Wnt ligands by PORCN is required for cell transport from the Golgi apparatus into the cell membrane for subsequent secretion into the extracellular space. Therefore, PORCN inhibitors are effective against tumors that display aberrant ligand-dependent Wnt/β-catenin activation [34]. Next, TNKSs belong to the superfamily of poly (ADP-ribose) polymerases (PARPs). Previous studies have identified TNKSs as potential therapeutic targets in cancer [35]. Although TNKSs play a role in several cellular functions [36], they are characterized as Wnt/β-catenin signaling modulators that PARylate and tag Axin1/2 (another member of the β-catenin destruction complex) for subsequent degradation [37]; therefore, TNKS inhibitors stabilize Axin1/2, increasing β-catenin degradation. Unlike PARP1 inhibitors (such as Olaparib or rucaparib) that have demonstrated their efficacy in several cancers [38], the evidence on TNKS inhibitors is somehow limited to preclinical studies [39], with a few first phases or ongoing clinical trials [40,41]. Interestingly, investigators have recently speculated on the therapeutic potential of PARP1 inhibitors in GC, even in combination with immunotherapy [42]. This will probably require the identification of homologous recombination-deficient (HRD) GC patients that harbor mutations in critical genes such as *PALB2*, *BRCA1/2*, *ATM*, *RAD51C*, or *ARID1A* [43]. Notably, our study found *ARID1A* levels were up-regulated in NR patients in our cohort (Figure 1b,c); we also found *ARID1A* mutations were present in a subset of chemo-immunotherapy-resistant patients (Figure 2b). Furthermore, studies have postulated that *ARID1A* deficiency might confer sensitivity to PARP inhibitors in cancer patients [44]. Another therapeutic alternative is CBP/β-catenin antagonists. These are small molecules that interfere in the β-catenin-CBP (transcriptional coactivator) interaction. Importantly, CBP/β-catenin antagonists may offer an alternative for patients who harbor tumors with ligand-independent aberrant Wnt/β-catenin activation [45]. To date, the most studied CBP/β-catenin antagonist is PRI-724; at least four clinical trials have sought to evaluate its safety and efficacy in advanced pancreatic carcinoma (phase I; NCT01764477), in advanced solid tumors (phase I; NCT01302405), in advanced myeloid malignancies (phase I/II: NCT01606579), or metastatic colorectal cancer (phase II: NCT02413853). Unfortunately, the phase I/II study on advanced myeloid malignancies is yet to publish its results, and phase Ib in advanced pancreatic adenocarcinoma reported its safety but a modest clinical efficacy as the second line in combination with gemcitabine [46]; the other trials were terminated early or withdrawn. For a complete list of Wnt signaling inhibitors, please see [47,48].

Finally, our study has certain limitations. First, it includes a relatively small number of patients; therefore, some of our findings should be further confirmed and validated in a larger cohort. Second, other relevant biomarkers could not be obtained from patient records, including HER2 or EBV status, and, therefore, we could not establish GC subtypes. Third, the 3′MACE technique allows basic quantification and sequencing (3′ end) of transcripts but cannot identify gene variants and mutations. Future studies should apply RNAseq or similar techniques to learn about these alterations, particularly in chemo-immunotherapy-resistant patients. Fourth, our studies were specifically centered on primary resistance. As explained, secondary (or acquired) resistance mechanisms may differ from those discussed above and may require other strategies such as liquid biopsy and longitudinal studies.

## 4. Materials and Methods

### 4.1. Patients

Initially, a total of 45 advanced GC patients were retrospectively enrolled into our study between March 2018 and January 2020. All patients were diagnosed and treated with first-line chemo-immunotherapy at the Centro del Cancer in the Red de Salud UC-CHRISTUS at Santiago, Chile. Inclusion criteria included histopathologic and radiologic confirmation of advanced disease, treatment with chemotherapy plus immune checkpoint inhibitor, aged ≥ 18 yr-old, availability of pretreatment endoscopic FFPE gastric tumor biopsies, and accessible clinical data including medical records, pathologic, and radiologic reports. Complete clinical datasets were available in 39 out of 45 patients (Table 1), and a total of 34 passed quality controls. According to iRECIST criteria [49], 16 patients were categorized as responders (n = 16) and 18 were non-responders (n = 18) at 12–16 weeks of treatment. Median follow-up for all patients was 18 months (range: 5–68 months). Overall survival (OS) was defined as the time-lapse from diagnosis until death either by GC (event) or by any other cause (censorship) or the last follow-up clinical visit (censorship). Descriptive survival analysis was performed using the Kaplan–Meier method. The differences between responders (R) and non-responders (NR) were estimated using the LogRank test.

**Table 1 ijms-24-00001-t001:** Basic clinical characteristics of patients of the study (n = 39).

Characteristic, Units	Value
Median age, yr. (range)	61 (34–83)
Gender, n (%)	
Male	30 (76.9)
Female	9 (23.1)
+Comorbidities, n (%)	21 (53.8)
+FH of gastric cancer, n (%)	7 (17.9)
Primary tumor location, n (%)	
Upper third	9 (23.1)
Middle third	22 (56.4)
Lower third	8 (20.5)
Lauren histotype, n (%)	
Intestinal	23 (59)
Diffuse	9 (23.1)
Mixed	5 (12.8)
N/A	2 (5.1)
+Signet ring cells, n (%)	14 (35.9)
Differentiation grade, n (%)	
Well differentiated	2 (5.1)
Moderately differentiated	17 (43.6)
Poorly differentiated	17 (43.6)
Undifferentiated	1 (2.6)
N/A	2 (5.1)
+Metastases sites, n (%)	
Lymph nodes	7 (17.9)
Bone	2 (5.1)
Liver	11 (28.2)
Peritoneum	12 (30.8)
Other	7 (17.9)
+Adenopathies, n (%)	
Perigastric	15 (38.5)
Retroperitoneal	17 (43.6)
Other	7 (17.9)
Immunotherapy response, n (%)	
Non-responder	19 (48.7)
Responder	20 (51.3)

Abbreviations: yr.: years, FH: family history, N/A: not available.

### 4.2. 3′Massive Analysis of cDNA Ends (3′MACE) and Pathway Enrichment Analysis

Total RNA was isolated from formalin-fixed paraffin-embedded (FFPE) endoscopic tissue biopsies using the QIAamp DNA FFPE tissue kit (Cat. No./ID: 56404, QIAGEN, Hilden, Germany) and quantified by Quant-iT RiboGreen RNA quantitation kit (Invitrogen, Waltham, MA, USA); 3′ MACE libraries were constructed using the MERCURIUS BRB-seq kit following the recommendations provided by the manufacturer (Alithea genomics). Briefly, mRNA was isolated from total RNA samples, reversed transcribed using barcoded oligo-dT primers, and converted into double-stranded full-length cDNAs. Then, cDNA libraries were prepared comprising unique dual index with unique molecular identifier (UMI) strategy and were sequenced in a HiSeq 2000 (Illumina, San Diego, CA, USA) with 100bp SE. Raw data were then analyzed with fastqc (https://www.bioinformatics.babraham.ac.uk/projects/fastqc/ Accessed on 16 September 2022) and aligned with STAR to hg38 version of the human genome (https://github.com/alexdobin/STAR Accessed on 16 September 2022) using the “solo” mode to create UMI count matrices for QC and data processing, respectively. Then, we used the BRB-seqTools to generate gene count matrices (https://github.com/DeplanckeLab/BRB-seqTools Accessed on 16 September 2022). Gene count matrices and clinical metadata were then uploaded and analyzed in Deseq2 [50] to identify differentially expressed genes (applying a *p*-value < 0.01 and log^2^foldChange > 0.58). Finally, pathway enrichment analyses were performed in the Protein ANalysis THrough Evolutionary Relationships (PANTHER) phylogenomic analysis tool [51].

### 4.3. Profiling by Next Generation Sequencing

Genomic DNA was isolated from FFPE biopsies of patients previously categorized as NR to chemo-immunotherapy (*n* = 4), as described above using the QIAamp DNA FFPE Tissue Kit (from QIAgen) according to the manufacturer’s protocol. Then, the samples were analyzed by commercially available NGS services with panels of cancer-related genes (BGI-Sentis or Foundation One). Clinically relevant alterations are summarized in Appendix A.

## 5. Conclusions

Our findings suggest that angiogenesis and the Wnt/β-catenin signaling pathways are enriched in advanced-stage GC patients with primary resistance to chemo-immunotherapy. Moreover, based on NGS data and pathway interactions (via crosstalk) described in the literature, we speculate that PI3K pathway mutations/alterations may play a role in treatment resistance. These pathways may offer actionable targets and alternative therapeutic strategies for advanced-stage GC patients that display primary resistance to the current standard of care.

## Figures and Tables

**Figure 2 ijms-24-00001-f002:**
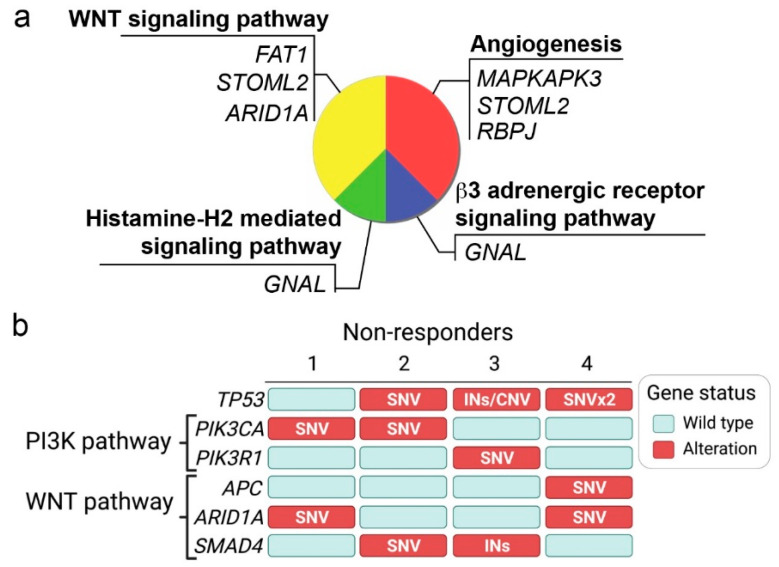
Enriched pathways in non-responders to chemo-immunotherapy. (**a**) Cell signaling pathway enrichment analyses were performed by obtaining gene lists of differentially expressed genes. (**b**) Diagram showing clinically relevant alterations in four chemo-immunotherapy NR advanced GC patients. Abbreviations: SNV: single nucleotide variant, INs: insertion, CNV: copy number variation.

**Figure 3 ijms-24-00001-f003:**
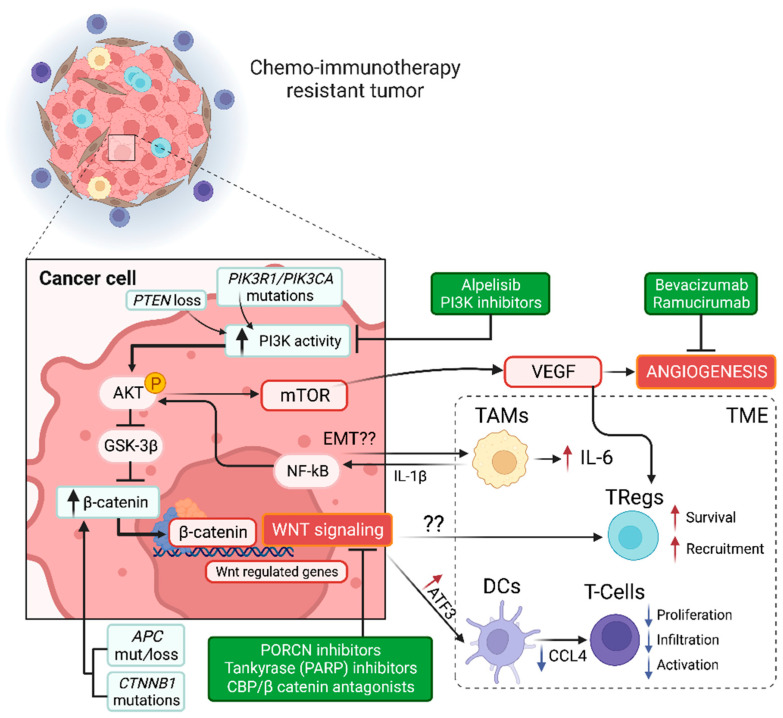
Hypothetical model of chemo-immunotherapy-resistant tumors in advanced GC patient and potential therapeutic interventions. Briefly, aberrant high PI3K activity caused by *PIK3CA*/*PIK3R1* mutations and/or *PTEN* loss increases Akt activation. In turn, activated Akt stimulates VEGF secretion and angiogenesis via mTOR. Additionally, Akt increases β-catenin activity and Wnt signaling via crosstalk with GSK3β. Increased Wnt signaling affects the tumor microenvironment (TME) via “immune exclusion”, increasing Treg survival and reducing CD8^+^T-cell proliferation, infiltration, and activation via a decrease in CCL4 in dendritic cells (DC). Highlighted potential therapeutic interventions include PI3K inhibitors such as Alpelisib, angiogenesis inhibitors such as bevacizumab or ramucirumab, and Wnt/β-catenin signaling inhibitors such as porcupine (PORCN) inhibitors, tankyrase inhibitors, or CBP/β-catenin antagonists.

## Data Availability

The data presented in this study are openly available in https://dataview.ncbi.nlm.nih.gov/object/PRJNA887186?reviewer=1c5visiumskog88qaojeifko0p Accessed on 16 September 2022.

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
