# Peer review of "Differentially Expressed Genes and Signaling Pathways Potentially Involved in Primary Resistance to Chemo-Immunotherapy in Advanced-Stage Gastric Cancer Patients"

_ijms, 2022, doi:10.3390/ijms24010001_

Round 1

Reviewer 1 Report

Overview and general recommendation:

In the manuscript, 45 advanced GC patients are included in this research. The patients are classified as responders and non-responders. 3’MACE analysis, pathway enrichment analyses and next generation sequencing show that genes of Wnt/β-catenin and the phosphoinositide 3-kinase (PI3K) pathway are differentially expressed.

Overall, the paper is OK. It is well written and the data are presented clearly. The authors performed detailed background research. All data are clearly organized and well documented. But I suggest the authors to re-organize the result and present it in a way which can show the logic of how you design your project.

Major comments:

1.      The sample amount is too small to draw a conclusion.

2.      I suggest the authors to re-organize the result. The result can be divided into 2 or 3 parts and the authors can present your work in a more logical way. They can introduce the how they start the work, why they employ 3’MACE analysis, pathway enrichment analyses and next generation sequencing in this research in detail.

3.      Overall, the research lacks significance and novelty. Many similar researches are conducted so the authors should include more data and more analysis to show why this research is more distinguished than other researches.

Author Response

Reviewer 1

Overview and general recommendation:

In the manuscript, 45 advanced GC patients are included in this research. The patients are classified as responders and non-responders. 3’MACE analysis, pathway enrichment analyses and next generation sequencing show that genes of Wnt/β-catenin and the phosphoinositide 3-kinase (PI3K) pathway are differentially expressed.

Overall, the paper is OK. It is well written and the data are presented clearly. The authors performed detailed background research. All data are clearly organized and well documented. But I suggest the authors to re-organize the result and present it in a way which can show the logic of how you design your project.

Major comments:

1-The sample amount is too small to draw a conclusion.

R1. First of all, we would like to thank you for taking the time to review our work. Regarding your comment, we acknowledge this is a small sample. However, given that the magnitude of the effect in our experiments is uncertain it is difficult to determine an appropriate sample size. In order to minimize uncertainty in our results we used samples that passed several quality controls and applied highly stringent p-values. Concomitantly, we have changed the title of our manuscript to be more cautious in our conclusions

2- I suggest the authors to re-organize the result. The result can be divided into 2 or 3 parts and the authors can present your work in a more logical way. They can introduce the how they start the work, why they employ 3’MACE analysis, pathway enrichment analyses and next generation sequencing in this research in detail.

R2. As requested by the reviewer we have reorganized our results and have divided the Results section of the revised manuscript into 3 sections adding subheadings to present our findings:

3.1. Selection of patients

3.2. Analysis of differentially expressed genes between R and NR

3.3. Analysis of enriched pathways and NGS in NR patients

We hope this is acceptable for the reviewer

3- Overall, the research lacks significance and novelty. Many similar researches are conducted so the authors should include more data and more analysis to show why this research is more distinguished than other researches.

R3. We disagree with the reviewer. Regarding its novelty, to our knowledge, there are no reports on the transcriptomic profile of chemo-immunotherapy responders versus non-responders aiming to identify differentially expressed genes associated with primary resistance. In addition, our study employs an innovative approach to analyze endoscopic biopsy samples; the 3’ MACE technique is a reliable lower-cost alternative to analyze differentially expressed genes in this type of sample (FFPE) and therefore this also potentiates the novelty of our work. Regarding its significance, our manuscript is intended to be part of a special issue on molecular oncology in Chile. Several studies suggest that gastric cancer displays a high geographical heterogeneity. Our study contributes to define the Chilean population of advanced gastric cancer patients, a country with high incidence and mortality associated with this disease.

Reviewer 2 Report

The study by Mauricio P. Pinto et al. is well conducted and presented. The topic is relevant and actual. The methods are correct. Results are logically and honestly presented. The Discussion is complete, perhaps a little redundant.

Given the predominantly descriptive nature of the article which does not report mechanistic data, I suggest to tone down the title and to be more sharp and consistent “Differential expression of genes involved in… are putatively associated with primary…”.

I have just two minor comments:

 “…nivolumab (a form of immunotherapy)” line 58, is too vague, please briefly specify the mechanism of action of nivolumab.

I would appreciate to see a supplementary table reporting the compliance to chemo-immunotherapies or, at least, how many patients eventually suffered or stopped the treatment for AEs or more specific irAEs (between Rs and NRs). This information would increase the strength of the article.

Author Response

 Reviewer 2

The study by Mauricio P. Pinto et al. is well conducted and presented. The topic is relevant and actual. The methods are correct. Results are logically and honestly presented. The Discussion is complete, perhaps a little redundant.

Given the predominantly descriptive nature of the article which does not report mechanistic data, I suggest to tone down the title and to be more sharp and consistent “Differential expression of genes involved in… are putatively associated with primary…”.

R- Thank for your suggestion. In response, we have modified the title of our manuscript as suggested

I have just two minor comments:

 “…nivolumab (a form of immunotherapy)” line 58, is too vague, please briefly specify the mechanism of action of nivolumab.

R1. OK. We have added a short paragraph describing the mechanism of action of nivolumab in the introduction section of the revised manuscript (second paragraph)

I would appreciate to see a supplementary table reporting the compliance to chemo-immunotherapies or, at least, how many patients eventually suffered or stopped the treatment for AEs or more specific irAEs (between Rs and NRs). This information would increase the strength of the article.

R2. Interesting point. Unfortunately, this was an observational study and our access to patient information was very limited and only provides basic data and response status (R or NR). The data regarding AEs and irAEs is usually available in clinical trials specifically designed as interventional studies, which is not our case here.

Round 2

Reviewer 1 Report

In this study, 45 advanced GC patients are analyzed by 3’MACE analysis, pathway enrichment analyses and next generation sequencing and the genetic characters are described. I find that the authors have put considerable effort into addressing the reports of the referees. As a result the paper is very much improved and I have no problem in recommending it for publication.